# Increased Waist Circumference after One-Year Is Associated with Poor Chewing Status

**DOI:** 10.3390/healthcare12131341

**Published:** 2024-07-05

**Authors:** Riku Yamazaki, Komei Iwai, Tetsuji Azuma, Takatoshi Yonenaga, Yasuyuki Sasai, Kazutoshi Watanabe, Akihiro Obora, Fumiko Deguchi, Takao Kojima, Wakako Tome, Noriyuki Kitai, Takaaki Tomofuji

**Affiliations:** 1Department of Orthodontics, School of Dentistry, Asahi University, 1851-1 Hozumi, Mizuho, Gifu 501-0296, Japan; rick0107@dent.asahi-u.ac.jp (R.Y.); wakakot@dent.asahi-u.ac.jp (W.T.); nkitai@dent.asahi-u.ac.jp (N.K.); 2Department of Community Oral Health, School of Dentistry, Asahi University, 1851-1 Hozumi, Mizuho, Gifu 501-0296, Japan; ko-mei@dent.asahi-u.ac.jp (K.I.); tetsuji@dent.asahi-u.ac.jp (T.A.); yone0730@dent.asahi-u.ac.jp (T.Y.); ysssi0718@dent.asahi-u.ac.jp (Y.S.); 3Human Health Center, Asahi University Hospital, 3-23 Hashimoto-cho, Gifu, Gifu 500–8523, Japan; watanabe@hosp.asahi-u.ac.jp (K.W.); akiobora18@hosp.asahi-u.ac.jp (A.O.); deguchi5757@yahoo.co.jp (F.D.); tkojima-gi@umin.ac.jp (T.K.)

**Keywords:** chewing, waist circumference, health checkups

## Abstract

Background: This study aimed to investigate the relationship between an increase in waist circumference (WC) after 1 year and self-reported chewing status in 10,870 Japanese adults who had received health checkups. Subsequently, 8068 participants were included in the final analysis. Methods: We defined an increase in WC ≥ 5 cm after 1 year as an unhealthy increase; in total, 613 (7.5%) respondents met this criterion. Chewing status was evaluated using a self-reported questionnaire at baseline; 1080 (13%) respondents were diagnosed with poor chewing status. Results: After adjusting for age, gender, WC, body mass index (BMI), and chewing status, an increase in WC ≥ 5 cm was found to be positively associated with gender (females: odds ratios [ORs]: 1.206; 95% confidence intervals [CIs]: 1.008–1.443), WC (ORs: 0.967; 95% CIs: 0.954–0.981), BMI (≥25 kg/m^2^; ORs: 2.194; 95% CIs: 1.715–2.808), and chewing status (poor; ORs: 1.356; 95% CIs: 1.084–1.697). Conclusions: These findings suggest that increased WC after 1 year was associated with self-reported poor chewing status in Japanese adults.

## 1. Introduction

Overweight and obesity are defined as abnormal or excessive fat accumulation, which is increasing in prevalence and is now considered to be a global epidemic. In addition, it has been reported that overweight and obesity are associated with several major diseases, such as cardiovascular disease, diabetes, and hypertension [1]. In 2019, in Japan, the National Health and Nutrition Examination Survey reported overweight or obesity in 33.0% and 22.3% of males and females aged ≥20 years, respectively [2]. In addition, people with overweight and obesity have been reported to have higher medical expenditures compared with those without them [3]. In light of this critical national health situation, in Japan, a national program for preventing obesity is considered very important.

Oral health is important because it not only helps maintain healthy teeth and gingiva but also contributes to general health. Several studies have focused on the risk of overweight and obesity from the viewpoint of oral health status, including chewing status. For instance, a systematic review revealed that poorer mastication was associated with overweight and obesity in 12 of 16 cross-sectional studies [4]. It is known that the number of opposing posterior teeth is related to overweight and obesity [5]. It was also shown that poor chewing status was associated with increased triglyceride (TG) levels [6], decreased high-density lipoprotein (HDL) cholesterol levels [6], and increased hemoglobin A1c (HbA1c) levels [7], which could be the risk of overweight and obesity [8,9,10,11]. Furthermore, it was reported that poor chewing may inhibit the production of histamine via the trigeminal nerve, inhibit the breakdown of visceral fat, and promote fat synthesis [12,13]. These findings indicate that chewing status is a potential rational factor related to overweight and obesity. However, to our knowledge, only a few studies have investigated the relationship between chewing status and the risk of being overweight/obese. Further clarification of this relationship could help promote the development of improved methods for obesity prevention.

Waist circumference (WC) is a simple method to assess abdominal adiposity that is easy to assess, and therefore, it is the most commonly accepted first step to determine an individual’s degree of overweight or obesity [14]. The measurement of WC has the advantage that it can also be used to assess overweight and obesity, including skinny obesity, and normal or less body weight [15]. Therefore, the International Atherosclerosis Society and the International Chair on Cardiometabolic Risk Working Group on Visceral Obesity have recommended the adoption of WC as a routine measurement in clinical practice to classify overweight/obesity [16]. Evaluating changes in WC could therefore be useful to clarify the relationship between chewing status and the risk of being overweight/obese.

In Japan, the Ministry of Health, Labour and Welfare requires medical insurers to conduct specific health checkups for insured persons aged 40–74 years [17]. These checkups focus on the accumulation of visceral fat and assume that a WC exceeding a certain standard value increases the risk of lifestyle-related diseases. Unfortunately, dental checkups are not included in standard health checkups. However, the medical questionnaire used in these health checkups includes an item on chewing status. These kinds of health checkups are a core health policy in Japan for more than 30 million people, and if it were to become clear that chewing status is involved in changes in WC, it would provide a new screening method for preventing overweight and obesity in Japan.

Specific health checkups are carried out once a year. If it is known at the time of the specific health checkup how an individual’s chewing status relates to changes in WC 1 year later, this fact can easily be introduced into health guidance for overweight/obesity prevention. Given this background, in the present study, we hypothesized that chewing status might influence an increase in WC after 1 year, and surveyed Japanese adults who received a specific health checkup.

## 2. Materials and Methods

### 2.1. Participants

All study participants received a specific health checkup at Asahi University Hospital Human Health Center between April 2018 and March 2019 (baseline). A total of 10,870 people were recruited, of whom, 799 were excluded because information on body mass index (BMI), WC, blood parameters, chewing status, or lifestyle habits was missing at baseline or after the 1-year follow-up visit. In addition, because people with BMI < 18.5 kg/m^2^ are defined as “under-weight” [18,19], and these individuals would not be considered unhealthy if their WC increased in the future, 1385 participants with BMI < 18.5 kg/m^2^ were excluded [20,21]. Accordingly, 8068 participants (5138 males and 2973 females; mean age, 51.0 years) were included in the final analysis between April 2019 and March 2020 (follow-up rate: 92.8%) (Figure 1). This study was approved by the Ethics Committee of Asahi University (No. 30018; approved 2018) and performed in accordance with the Declaration of Helsinki. All participants provided informed consent. This study followed the STROBE guidelines.

### 2.2. Assessment of BMI and WC

Trained nurses measured each participant’s height (in centimeters to the nearest 0.1 cm) and body weight (in kilograms to the nearest 0.1 kg). BMI was calculated as weight divided by height in meters squared (kg/m^2^). WC (in centimeters to the nearest 0.5 cm) was measured horizontally at the umbilical level at the end of normal expiration [22,23]. Measurements of WC were taken twice, and the average of two values was used.

### 2.3. Measurement of Blood Pressure

Blood pressure was measured using a standard sphygmomanometer (HBP-9030; OMRON, Kyoto, Japan), with the participants sitting on chairs after a minimum of 5 min of rest [24,25]. Measurements of blood pressure were taken twice by the nurse, and the average of two values was used.

### 2.4. Measurement of Blood Parameters

Venous fasting blood samples were collected, and serum levels of TG, HDL cholesterol, and HbA1c were determined using an automatic analyzer (Dimension Vista 1500; Siemens Healthiness Japan, Tokyo, Japan; DM-JACK; Kyowa Medix, Tokyo, Japan) [26,27].

### 2.5. Evaluation of Chewing Status

To evaluate chewing status, we used the standard self-administered questionnaire mandated by the Act on Assurance of Medical Care for Elderly in Japan. The responses to the relevant questionnaire items were as follows: “I can eat anything”, “Sometimes it is difficult to chew because of dental problems such as dental caries and periodontal disease”, and “I can hardly chew” [28]. Respondents who answered “Sometimes it is difficult to chew because of dental problems such as dental caries and periodontal disease” or “I can hardly chew” were considered to have poor chewing status [29].

### 2.6. Other Items Surveyed from the Self-Administered Questionnaire

The self-administered questionnaire was used to collect information on the participants’ gender, age, smoking habits, amount of drinking, exercise habits, and sleep status. Smoking habits were classified as being present if a participant smoked at least one cigarette per day [30]. Participants who drank > 180 mL of sake, >500 mL of beer, >80 mL of shochu, >60 mL of whiskey double, or >240 mL of wine were defined as heavy drinkers [31]. Participants who exercised ≥30 min ≥2 days/week were defined as having a regular exercise habit [29]. Sleep status was categorized as poor or good [32]. These were validated and standardized items used in the specific health checkups.

### 2.7. Statistical Analysis

The normality of the data was confirmed using Kolmogorov–Smirnov tests. Because all continuous variables were not normally distributed, non-parametric tests were used in our study. The outcome of interest was the 1-year change in WC from baseline to the follow-up visit (referred to as a change in WC). Changes in WC were calculated by subtracting WC at baseline from WC at follow-up. A 5-cm increase in WC is associated with increased cardiovascular risk factors [33]. Therefore, in this study, an increase in WC of ≥5 cm was defined as an unhealthy change. Continuous variables (age, blood pressure, HbA1c, TG, HDL cholesterol, and WC) were presented as the median (Quartile 1 and Quartile 3). Differences between the two groups were evaluated using the Mann–Whitney U test for continuous variables and the chi-square test for categorical variables out of non-parametric tests. In the multivariate logistic regression, variables with *p*-values > 0.05 were excluded from the model, and the third category of variables related to the sample (age, gender, WC, and chewing status) and variables with significant differences in the univariate logistic analysis were adjusted for in these analyses. All data were analyzed using SPSS (version 27; IBM Japan, Tokyo, Japan). All *p*-values < 0.05 were considered statistically significant.

## 3. Results

Table 1 shows the baseline characteristics of the participants with a chewing status of poor or good. In this study, 1080 participants (13%) were diagnosed with poor chewing status. Participants with poor chewing status were significantly older (*p* < 0.001) and had higher HbA1c (*p* = 0.004) and TG levels (*p* = 0.013) than those with good chewing status. Participants with poor chewing status also had significantly lower levels of HDL cholesterol (*p* < 0.001) than those with good chewing status. Furthermore, a significantly higher proportion of participants with poor compared with good chewing status had a smoking habit (*p* < 0.001), were heavy drinkers (*p* = 0.027), had regular exercise habits (*p* = 0.003), and had poor sleeping status (*p* < 0.001).

Table 2 shows the results of a comparison of WC by chewing status at baseline and after 1 year. The median (Quartile 1 and Quartile 3) WC in the good chewing status group was 80.0 cm (75.0, 86.0) at both baseline and after 1 year. By contrast, the median (Quartile 1 and Quartile 3) WC in the poor chewing status group was 80.0 cm (75.0, 87.0) at baseline and 81.5 cm (76.0, 87.5) after 1 year. The WC values in the poor chewing status group were significantly higher than those in the good chewing status group at both baseline (*p* = 0.017) and after 1 year (*p* < 0.001). 

Table 3 shows the characteristics of changes in WC after 1 year by chewing status. Of the participants with good chewing status, 510 (7%) had an increase in WC of ≥5 cm after 1 year, whereas, among those with poor chewing status, 103 (10%) had an increase in WC of ≥5 cm after 1 year. The proportion of participants with changing WC ≥ 5 cm in the poor chewing status group was significantly higher than those in the good chewing status group (*p* = 0.010).

Table 4 shows the crude odds ratios (ORs) and 95% confidence intervals (CIs) for an increase in WC of ≥5 cm after 1 year. The results showed that the risk of a ≥ 5 cm increase in WC after 1 year was significantly correlated with gender (females: ORs: 1.264; 95% CIs: 1.069–1.494), BMI (≥25 kg/m^2^: ORs: 1.369; 95% CIs: 1.148–1.632), and chewing status (poor: ORs: 1.339; 95% CIs: 1.072–1.672) at baseline.

Table 5 shows the adjusted ORs and 95% CIs for a ≥ 5 cm increase in WC after 1 year. After adjusting for age, gender, WC, BMI, and chewing status, the risk of an increase in WC of ≥5 cm after 1 year was significantly correlated with gender (females: ORs: 1.206; 95% CIs: 1.008–1.443), WC (ORs: 0.967; 95% CIs: 0.954–0.981), BMI (≥25 kg/m^2^: ORs: 2.194; 95% CIs: 1.715–2.808), and chewing status (poor: ORs: 1.356; 95% CIs: 1.084–1.697) at baseline.

## 4. Discussion

To our knowledge, this is the first study to examine the association between increased WC after 1 year and chewing status. The median WC in the poor chewing status group was higher than that in the good chewing status group at both baseline and follow-up. The results of the multivariate logistic regression showed that a ≥ 5 cm increase in WC after 1 year was associated with poor chewing status at baseline (ORs: 1.356; 95% CIs: 1.084–1.697) after adjusting for age, gender, WC, and BMI. A previous study reported that a ≥ 5 cm increase in WC was associated with the development of cardiovascular disease [1], suggesting that poor chewing status is a risk factor for a future unhealthy increase in WC. Furthermore, it is accepted that an increase in WC reflects an increase in visceral fat [14]. The present results indicate that poor chewing status could be a risk factor for an unhealthy increase in visceral fat.

Several studies have examined the relationships among chewing status, weight change, and BMI. For instance, a cross-sectional study showed that the categorical chewing number was related independently to body weight increments of >10 kg from 20 years of age in persons aged 35–61 years [34]. Another cross-sectional study reported statistically significant negative correlations between BMI and the number of chewing cycles (r = −0.296, *p* = 0.020) and chewing duration (r = −0.354, *p* = 0.005) in fully dentate healthy adults [35]. The other clinical study also showed that masticatory factors evaluated with a wearable device were independently associated with BMI [36]. These findings suggest that chewing status is associated with a risk for overweight/obesity via changes in weight and BMI. In the present study, poor chewing status was associated with an unhealthy increase in WC. Therefore, the previous and present studies all support the concept that chewing status is a target factor for overweight/obesity prevention.

Specified medical checkups are conducted once a year in Japan. The present study was a 1-year study conducted to evaluate changes until the next specified medical checkup. In other words, the results of this study indicated that respondents with poor mastication status in the specified medical checkups were at high risk of unhealthy increases in WC by the next specified medical checkup. Therefore, in order to prevent unhealthy accumulation of visceral fat, it is considered necessary not to leave participants who responded with poor chewing status until the next specified medical checkup. Self-reported chewing status has been shown to be associated with the number of existing teeth, the occlusal status of the molars, and the number of functional teeth [37]. Therefore, dental health guidance for respondents with poor chewing status may be important in terms of preventing unhealthy increases in WC.

Several possible mechanisms may explain the association between chewing status and increases in WC. People with poor compared with good chewing status tend to consume fewer fruits and vegetables and more high-energy foods, which may lead to unhealthy increases in WC [38]. Then, it has been reported that chewing promotion increases the partial pressure of oxygen in the brain and activates brain function, thereby increasing the number of calories consumed in the body [39,40]. Furthermore, it has been demonstrated that overweight participants chewed less and ingested more calories in a randomized cross-over design study [41]. Therefore, the participants with poor chewing status in the present study may have had a reduced total number of calories consumed in the body, which may be associated with future increases in WC. 

The other study reported that chewing problems increase the likelihood of both low BMI and weight loss [42]. This finding differs from the concepts of our study, in which poor chewing status is a risk for increased WC. The subjects in the previous study were nursing home residents [42], and the subjects in this study were health checkup recipients. The association between chewing status and increased WC may vary depending on the characteristics of the population studied. 

In the present study, a significant association was found between increases in WC and female gender. This finding is consistent with previous studies reporting that females accumulate fat more readily than males [43,44]. The reason that females are more likely to accumulate fat than males may be related to their lower basal metabolic rate and the influence of female hormones [43,44].

On the other hand, no association was found between the increase in WC and age in this study. By contrast, a previous study found an association between increased WC and older age [45]. This discrepancy may be related to the difference in the age distribution of the participants in the previous and present studies; the 2019 National Health and Nutrition Survey reported that the rate of increase in WC was significantly higher among those aged ≥60 years [2]. The participants in another previous study [45] were older than those in the present study, with a mean age of 65 years (vs. 51.0 years in the present study).

In Japan, a BMI ≥ 25 kg/m^2^ is considered as an indicator of obesity [46]. Therefore, the present findings suggest that participants with obesity are more likely to have an increased WC after 1 year. This supports a previous study showing that participants with a higher BMI were more likely to have an increased WC [47].

Our study implies that the assessment of chewing status by a self-administered questionnaire proved to be a useful predictor of unhealthy accumulation of visceral fat. In other words, our study suggests that the assessment of the oral cavity in a self-administered questionnaire at health checkups is important for considering future health status.

This study has several limitations. First, socioeconomic factors such as the participants’ income and educational levels were not considered. Socioeconomic factors are reportedly involved in unhealthy conditions such as increases in WC [48]; however, these were unclear in the present study. Second, this study included only participants who received a specific health checkup at Asahi University Hospital Human Health Center. According to the 2018 National Health and Nutrition Survey, the proportion of people with obesity in their 50s was 25.9% [2], which is higher than the proportion of participants in the present study (20.2%). Therefore, results may differ for different health populations, and external validity should be considered. Third, there is no data on dietary intake because this item is not included in the specific health checkup questionnaire. A previous review has indicated that the optimal diet for the prevention of weight gain and obesity is low in fat and sugar-rich beverages and high in carbohydrates, fiber, grains, and protein [49], so adding data on dietary intake to the analysis would improve the reliability of the results of our study. On the other hands, we have the data on the frequency of intake of beef/pork, poultry, fish, dairy products, sweet foods, and vegetables in the daily diet, and tried to analysis these data. As a result of the analysis, there were no significant differences between an increase in WC ≥ 5 cm and these data (Appendix A). Finally, there was a significant difference in age between the good chewing status group and the poor chewing status group (Table 1), and this could be a bias. However, our multivariate analysis showed an association between poor chewing status and increased WC independent of age (Table 5). On the other hand, a major strength of the present study is its sample size of more than 8000 Japanese individuals. This sample size should be sufficient to show the relationship between chewing status and future increases in WC, which may be useful for inferring factors that contribute to the unhealthy accumulation of visceral fat in Japanese adults.

## 5. Conclusions

The results of the present study revealed an association between increased WC ≥ 5 cm after 1 year and self-reported poor chewing status. Poor chewing status may be an important risk factor indicating the unhealthy accumulation of visceral fat in Japanese adults. Therefore, recommending that examinees who answered with poor chewing status in a specific health checkup see a dentist may contribute to preventing the unhealthy accumulation of visceral fat in the future, but further research is needed.

## Figures and Tables

**Figure 1 healthcare-12-01341-f001:**
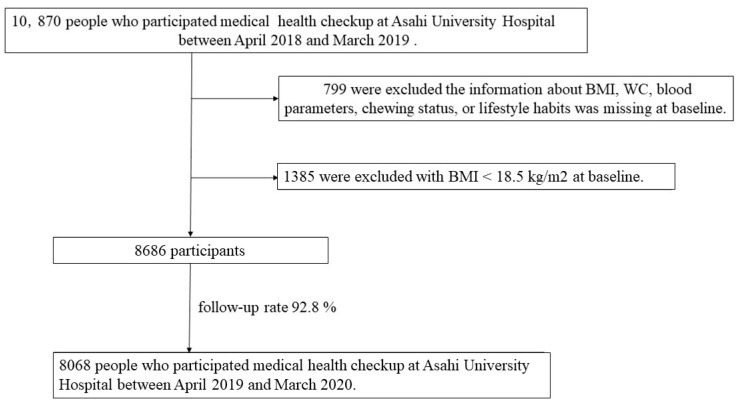
Flowchart in this study.

**Table 1 healthcare-12-01341-t001:** Baseline characteristics of the participants by chewing status.

Factor	Chewing Status		*p*-Value
	Good (n = 6988)	Poor (n = 1080)	
Age (year)	51 (45, 56)	53 (47, 58)	<0.001 *
Gender (female) (n [%])	2591 (37%)	339 (31%)	0.131
Maximal blood pressure (mmHg)	121 (111, 131)	122 (112, 131)	0.171
Diastolic blood pressure (mmHg)	76 (68, 84)	77 (68, 84)	0.158
BMI 25 ≥ (kg/m^2^) (n [%])	1898 (27%)	297 (28%)	0.816
HbA1c (%)	5.4 (5.3, 5.6)	5.4 (5.3, 5.7)	0.004 *
TG (mg/dL)	70 (49, 103)	75 (51, 107)	0.013 *
HDL cholesterol (mg/dL)	60 (50, 73)	58 (48, 71)	<0.001 *
Presence of smoking habits (n [%])	1099 (16%)	248 (23%)	<0.001 *
Heavy drinking (n [%])	813 (12%)	151 (14%)	0.027 *
Presence of regular exercise habits (n [%])	5521 (79%)	896 (83%)	0.003 *
Poor sleeping status (n [%])	2229 (32%)	478 (44%)	<0.001 *

Abbreviations: BMI, body mass index; HbA1c, hemoglobin A1c; TG, triglycerides; WC, waist circumference. * *p* < 0.05, using the chi-square test or Mann–Whitney U test.

**Table 2 healthcare-12-01341-t002:** Comparison of WC by chewing status at baseline and follow-up.

Factor	Chewing Status		*p*-Value
	Good (n = 6988)	Poor (n = 1080)	
WC at baseline (cm)	80.0 (75.0, 86.0)	80.0 (75.0, 87.0)	0.017 *
WC at follow-up (cm)	80.0 (75.0, 86.0)	81.5 (76.0, 87.5)	<0.001 *

Abbreviations: WC, waist circumference. * *p* < 0.05, using the Mann–Whitney U test.

**Table 3 healthcare-12-01341-t003:** Characteristics of the values or changes in WC after 1 year by chewing status.

Factor	Chewing Status		*p*-Value
	Good n = 6988	Poor n = 1080	
Changing WC (cm) (n [%])			
−5 < a	245 (4%)	30 (3%)	0.220
−5 ≤ a < −2.5	714 (10%)	91 (8%)	0.068
−2.5 ≤ a < 2.5	4416 (63%)	670 (62%)	0.463
2.5 ≤ a < 5	1103 (16%)	186 (17%)	0.230
5 ≤ a	510 (7%)	103 (10%)	0.010 *

Abbreviations: WC, waist circumference. * *p* < 0.05, using the chi-square test. a: Amount of changing WC among 1 year.

**Table 4 healthcare-12-01341-t004:** Crude ORs and 95% CIs for an increase in WC of ≥5 cm after 1 year.

Factor		ORs	95% CIs	*p*-Value
Age (year)		1.008	0.997–1.019	0.146
Gender	Male	1	(reference)	0.006 *
	Female	1.264	1.069–1.494	
Smoking habits	Absence	1	(reference)	0.154
Maximal blood pressure (mmHg)		1.001	0.996–1.006	0.702
Diastolic blood pressure (mmHg)		1.001	0.994–1.008	0.714
HbA1c (%)		0.922	0.780–1.088	0.336
TG (mg/dL)		1.000	0.998–1.001	0.704
HDL cholesterol (mg/dL)		1.002	0.997–1.007	0.466
WC (cm)		0.993	0.983–1.002	0.123
BMI (kg/m^2^)	25 ≥	1	(reference)	<0.001 *
	>25	1.369	1.148–1.632	
Smoking habits	Absence	1	(reference)	0.154
	Presence	1.166	0.944–1.442	
Amount of drinking	Not heavy	1	(reference)	0.771
	Heavy	0.963	0.744–1.245	
Regular exercise habits	Presence	1	(reference)	0.326
	Absence	1.111	0.901–1.373	
Sleeping status	Good	1	(reference)	0.314
	Poor	0.915	0.770–1.087	
Chewing status	Good	1	(reference)	0.010 *
	Poor	1.339	1.072–1.672	

Abbreviations: WC, waist circumference; BMI, body mass index; HbA1c, hemoglobin A1c; TG, triglycerides; ORs, odds ratios; CIs, confidence intervals. * *p* < 0.05, using univariate logistic regression analysis.

**Table 5 healthcare-12-01341-t005:** Adjusted ORs and 95% CIs for an increase in WC of ≥5 cm after 1.

Factor		ORs	95% CIs	*p*-Value
Age (year)		1.011	1.000–1.023	0.050
Gender	Male	1	(reference)	0.041
	Female	1.206	1.008–1.443	
WC (cm)		0.967	0.954–0.981	<0.001 *
BMI (kg/m^2^)	25>	1	(reference)	<0.001 *
	>25	2.194	1.715–2.808	
Chewing status	Good	1	(reference)	0.008 *
	Poor	1.356	1.084–1.697	

Abbreviations: WC, waist circumference; BMI, body mass index; ORs, odds ratios; CIs, confidence intervals. * *p* < 0.05, using multivariate logistic regression analysis. Adjustment for age, gender, BMI, and chewing status. Hosmer–Lemeshow fit test; *p* = 0.923.

## Data Availability

The data presented in this study are available on request from the corresponding author. The data are not publicly available due to ethical restrictions.

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
