# Peer review of "Increased Waist Circumference after One-Year Is Associated with Poor Chewing Status"

_healthcare, 2024, doi:10.3390/healthcare12131341_

Round 1

Reviewer 1 Report

Comments and Suggestions for Authors

Comments to the author:

The authors aim to investigate the relationship between self-reported chewing status and increases in waist circumference (WC) after 1 year in 10,870 Japanese adults who had received health checkups. Congratulate the authors for the work done, which is very interesting and contains valuable information. I have the following suggestions:

Results:

-       There is no data regarding dietary intake of participant and their effects on the results. It is very important and effective factor.

-       Please add the statistical methods and the p-values that considered statistically significant in the tables 3, 4 and 5 captions.

Author Response

There is no data regarding dietary intake of participant and their effects on the results. It is very important and effective factor.

Response: We thank the reviewer for your valuable advice. Although we are agreement with your comments, we do not have the data about dietary intake of participant. We have emphasized this limitation in the discussion (lines 322-326). On the other hands, we have the data on the frequency of intake of beef/pork, poultry, fish, dairy products, sweet foods, and vegetables in the daily diet, and tried to analysis these data. As a result of the analysis, there were no significant differences between an increase in WC ≥ 5 cm and these data. Please see the supplemental table.

Please add the statistical methods and the p-values that considered statistically significant in the tables 3, 4 and 5 captions.

Response: We thank the reviewers for your valuable advice. We have revised our manuscript according to your suggestion (Table 3-5 captions). In addition, we have added the sentence (lines 179-180).

Reviewer 2 Report

Comments and Suggestions for Authors

Through this study, you provided insight on the association between the poor chewing status and the changed in waist circumference.

However, I think there are some things that need to be revised in presenting the results on the subject of this study.

First, it seems unreasonable to describe a comparative study with a one-year term as a longitudinal study. Longitudinal studies employ continuous or repeated measures to follow particular individuals over prolonged periods of time—often years or decades.

Second, the subject of this study is to present the association between poor chewing status and waist circumference, and it is necessary to clearly describe the relationship between overweight and obesity, and an increased in waist circumference in the introduction. I think that the concept of the overweight or the increased in waist circumference is viewed as the same. In the case of skinny obesity, normal or less body weight, but there is abdominal obesity. Therefore, I think it is necessary to clarify the concept of this.

Third, it seems that the reason or rationale for the selection of triglyceride (TG), HDL cholesterol, and hemoglobin A1c (HbA1c) as variables should be presented in the introduction.

Finally, when looking at the statistical analysis results of this study, it seems to be an analysis study of the influencing factors of the increased waist circumference.

Thank you for your efforts.

Author Response

First, it seems unreasonable to describe a comparative study with a one-year term as a longitudinal study. Longitudinal studies employ continuous or repeated measures to follow particular individuals over prolonged periods of time—often years or decades.

Response: We thank the reviewer for your valuable comments. As you pointed out, the one-year observation period may be too short for a longitudinal study. Therefore, we have changed the text so as not to use the term "longitudinal study" in the text. On the other hands, because there is no doubt that the same person was measured twice in this study, the wording of baseline and follow-up were retained. We have changed the title and revised the words (lines 2, 15, 49, 58-59, 272-273, 335-336).

Second, the subject of this study is to present the association between poor chewing status and waist circumference, and it is necessary to clearly describe the relationship between overweight and obesity, and an increased in waist circumference in the introduction. I think that the concept of the overweight or the increased in waist circumference is viewed as the same. In the case of skinny obesity, normal or less body weight, but there is abdominal obesity. Therefore, I think it is necessary to clarify the concept of this.

Response: We thank the reviewer for your valuable advice. We have added the sentence in the text (line 53-54, References 15).

Third, it seems that the reason or rationale for the selection of triglyceride (TG), HDL cholesterol, and hemoglobin A1c (HbA1c) as variables should be presented in the introduction.

Response: We thank the reviewer for your valuable advice. We have added the text in “Introduction” about rationale for the selection of TG, HDL cholesterol, and HbA1c as variables according to your suggestion (line 42-44, References 6-11).

Finally, when looking at the statistical analysis results of this study, it seems to be an analysis study of the influencing factors of the increased waist circumference.

Response: We thank the reviewer for your valuable comment. As you indicated, our study aimed to explore the influencing factors of the increased waist circumference. We have revised the text (lines 2, 24-25, 71-73, 251-252, 335-336).

Reviewer 3 Report

Comments and Suggestions for Authors

Thank you very much for the possibility to review this manuscript. I appreciate your work, however, I need to address some suggestions to improve its quality.

First of all, please double-check the punctuation, spelling, and style.

Introduction:

Please better describe the rationale of the study.

Material and Methods:

Please improve the Method section and better describe the research procedures. Why did you exclude individuals with a BMI below 18.5? Could this affect the study results? How many assessments of WC and blood pressure were made to obtain the final result taken to the analysis? What about the self-administered questionnaire? Is it a validated and standardized tool? Please discuss it in the main text.

Also, how did you assess the distribution of the data in the Statistical analysis section? Instead of 25 and 75 percentiles, I recommend using the Quartile 1 and 3 nomenclature. Please better describe the decision-making process in this section.

Results:

You should not start with Table 1 if it is way below the text. The individuals with poor chewing status were, in fact, older compared to those with good status. This could affect the study results. Please point it out as a limitation.

Table 3 - are those results statistically significant? Did you use the Chi-square test?

Discussion:

Please improve this section by adding more references, going deeper into the analysis, and better discussing the similarities and differences. Also better discuss your findings.

Limitations: I believe there are more limitations to confess. Please add age differences between groups and unequal sizes of the groups as your limitations.

Please add the practical implications of this study.

Conclusions: Please improve this section. What are the future directions in this research area?

References: Way below the expectations. Please add at least 10-15 more references. Please change the font.

Comments on the Quality of English Language

I found some punctuation, spelling and style errors

Author Response

First of all, please double-check the punctuation, spelling, and style.

Response: We thank the reviewer for your valuable advice. We have re-checked and revised the text.

Introduction: Please better describe the rationale of the study.

Response: We thank the reviewer for your valuable advice. We have revised the sentences in the Introduction section (lines 42-47, 49, 53-54, 58, 71-73).

Material and Methods: Please improve the Method section and better describe the research procedures. Why did you exclude individuals with a BMI below 18.5? Could this affect the study results?

Response: We thank the reviewers for your valuable advice. As you pointed out, participants with BMI < 18.5 kg/m2 were excluded in our study. Because the people with BMI < 18.5 kg/m2 is defined as "under-weight" in Japan and these individuals would not be considered unhealthy if their WC increased in the future. We have added the sentence (lines 81-83, References 18-21). In addition, adding people with BMI < 18.5 kg/m2 to the analysis did not change our conclusion that poor chewing status was associated with increased WC.

How many assessments of WC and blood pressure were made to obtain the final result taken to the analysis?

Response: We thank the reviewers for your valuable advice. Measurements of WC and blood pressure were taken twice, and the average of two values was used. We added the text (lines 98-99, 104-105).

What about the self-administered questionnaire? Is it a validated and standardized tool? Please discuss it in the main text.

Response: We thank the reviewers for your valuable advice. We used the items for self-administered questionnaire mandated by Act on Assurance of Medical Care for Elderly in Japan. These were validated and standardized items. We have emphasized these points in the text (lines 113-114, 127-128).

Also, how did you assess the distribution of the data in the Statistical analysis section?

Response: We thank the reviewers for your valuable advice. The normality of the data was confirmed using Kolmogorov-Smirnov tests. Because all continuous variables were not normally distributed, non-parametric tests (Mann–Whitney U test and the chi-square test) were used in our study. We have added the sentences (lines 131-132, 139-140).

Instead of 25 and 75 percentiles, I recommend using the Quartile 1 and 3 nomenclature. Please better describe the decision-making process in this section.

Response: We thank the reviewers for your valuable advice. We have changed the words from “25 and 75 percentiles” to “Quartile 1 and 3” (lines 138, 158-159). Furthermore, we have added the text about decision-making process according to your suggestion (lines 131-132, 139-140, References 37).

Results: You should not start with Table 1 if it is way below the text.

Response: We thank the reviewers for your valuable advice. We have revised “Table” position correctly (Table 1).

The individuals with poor chewing status were, in fact, older compared to those with good status. This could affect the study results. Please point it out as a limitation.

Response: We thank the reviewers for your valuable advice. We have added the sentences to the limitations according to your suggestion (lines 326-329).

Table 3 - are those results statistically significant? Did you use the Chi-square test?

Response: We thank the reviewers for your valuable advice. We have used chi-square test. We have added the data in Table 3. In addition, because proportion of participants with changing WC ≥ 5 cm in poor chewing status group were significantly higher than those in good chewing status group (p = 0.010), we have added the sentence (lines 179-180).

Discussion: Please improve this section by adding more references, going deeper into the analysis, and better discussing the similarities and differences. Also better discuss your findings.

Response: We thank the reviewers for their valuable advice. We have revised the Discussion section (lines 266-267, 287-288, 291-295, 311-314, 322-329, References 36, 41, 42, 49).

Limitations: I believe there are more limitations to confess. Please add age differences between groups and unequal sizes of the groups as your limitations.

Response: We thank the reviewers for their valuable advice. We have added the sentence in our limitation (lines 322-326). Table 1 shows a significant difference in age between the good chewing status group and the poor chewing status group, but multivariate analysis shows an association between poor chewing status and increased WC independent of age (Table 5) (lines 326-329). We also believe that the unequal sizes between groups is not a problem based on the goodness of fit of the logistic analysis model.

Please add the practical implications of this study.

Response: We thank the reviewers for their valuable advice. The implication of our study is that the assessment of chewing status by a self-administered questionnaire proved to be a useful predictor of unhealthy accumulation of visceral fat. In other words, our study suggests that the assessment of oral cavity in a self-administered questionnaire at health checkup is important for considering future health status. We have added the sentences according to your suggestion (lines 314-317).

Conclusions: Please improve this section. What are the future directions in this research area?

Response: We thank the reviewers for their valuable advice. As a future development and directions in this research area, recommending that examinees who answered with poor chewing status in a specific health checkup see a dentist may contribute to preventing unhealthy accumulation of visceral fat in the future. We have added the sentence according to your suggestion (lines 337-340).

References: Way below the expectations. Please add at least 10-15 more references.

Response: We thank the reviewer for their valuable advice. We have added references according to your suggestion (References 6-13, 15, 18, 19, 21, 36, 41, 42, 49).

Please change the font.

Response: We thank the reviewer for your valuable advice. We have revised font correctly (Unified the font to Times New Roman).

I found some punctuation, spelling and style errors.

Response: We thank the reviewer for your valuable advice. We have re-checked and revised some punctuation, spelling and style correctly.

Round 2

Reviewer 2 Report

Comments and Suggestions for Authors

You have made a good correction reflecting the recommendations. Have a nice summer!

Reviewer 3 Report

Comments and Suggestions for Authors

Thank you for all improvements.

Comments on the Quality of English Language

It is improved